# Improving Fruit and Vegetable Accessibility, Purchasing, and Consumption to Advance Nutrition Security and Health Equity in the United States

**DOI:** 10.3390/ijerph191811220

**Published:** 2022-09-07

**Authors:** Bailey Houghtaling, Matthew Greene, Kaustubh V. Parab, Chelsea R. Singleton

**Affiliations:** 1Gretchen Swanson Center for Nutrition, Omaha, NE 68514, USA; 2School of Nutrition and Food Sciences, Louisiana State University (LSU) & LSU Agricultural Center, Baton Rouge, LA 70803, USA; 3Department of Human Nutrition, Foods, and Exercise, Virginia Tech, Blacksburg, VA 24061, USA; 4Department of Kinesiology and Community Health, University of Illinois at Urbana-Champaign, Champaign, IL 61820, USA; 5Department of Social, Behavioral, and Population Sciences, Tulane School of Public Health and Tropical Medicine, New Orleans, LA 70112, USA

**Keywords:** fruits and vegetables, diet, food accessibility, nutrition security, health equity

## Abstract

In recent years, national and local efforts to improve diet and health in the United States have stressed the importance of nutrition security, which emphasizes consistent access to foods and beverages that promote health and prevent disease among all individuals. At the core of this endeavor is fruit and vegetable (FV) consumption, a dietary practice that is integral to attaining and sustaining a healthy diet. Unfortunately, significant inequities in FV accessibility, purchasing, and consumption exist, particularly among populations that are socially and economically disadvantaged. To achieve nutrition and health equity in the United States, the field must center the goal of nutrition security and initiatives that aim to increase FV consumption, specifically, in future work. The *International Journal of Environmental Research and Public Health* (IJERPH) Special Issue titled “*Nutrition and Health Equity: Revisiting the Importance of Fruit and Vegetable Availability, Purchasing, and Consumption*” features several scholarly publications from experts conducting timely research on these topics. In this commentary, we (*1*) summarize the U.S.-based literature on inequities in FV accessibility, purchasing, and consumption, (*2*) describe how the contributions to this IJERPH special issue can advance nutrition security and health equity, and (*3*) outline future research questions from our perspective.

## 1. Introduction

The multifaceted relationship between diet and health continues to be a major public health concern. With cigarette smoking rates on the decline [1], poor diet is now the most prevalent modifiable risk factor for chronic disease in the United States (U.S.) [2]. Consuming a healthy diet, defined by the 2020–2025 Dietary Guidelines for Americans (DGA) as a diverse array of fruits and vegetables (FVs), whole grains, lean and plant-based proteins, and items lower in saturated fat, added sugars, and sodium [3], can reduce an individual’s risk for hypertension, type 2 diabetes, cardiovascular disease, and some cancers [2,4]. Nevertheless, data have persistently shown that most Americans’ diets fall short of national recommendations [3,5]. 

What is more concerning are the persistent inequities in dietary intake [6,7], which highlight barriers to the availability, affordability, and convenience of obtaining healthy foods and beverages among certain populations [8]. Concerned with these inequities, the United States Department of Agriculture (USDA) outlined plans to also pioneer nutrition security efforts in America in March 2022 [9,10]. Nutrition security is defined as consistent household accessibility (availability, affordability) to dietary products that “promote well-being and prevent (and if needed, treat) disease [9,10]”. This concept is noteworthy given a sole focus for decades on food security [11], which aims to ensure households have access to enough food, but does not necessarily emphasize nutritional quality. The concept of nutrition security also places structural barriers to healthy diets at the forefront of the conversation, a recognition that is required to advance research, practice, and policy capable of achieving nutrition and health equity [9,10]. 

At the center of this challenge is FV consumption, which deserves a unique emphasis regarding strategies to improve nutrition security and health equity. The adequate availability and consumption of culturally appropriate FVs is vital to achieving a healthy diet and preventing chronic disease [12]. Unfortunately, mirroring other nutrition inequities, FV consumption among adults and children are disproportionately lower among socially/economically disadvantaged and historically resilient populations, including people with lower incomes, racial and ethnic minority groups, certain members of the LGBTQIA2+ community [13,14], and people residing in rural areas [15,16,17,18,19]. 

Public consciousness of structural barriers to health have increased in recent years [20,21,22,23], primarily magnified by the COVID-19 pandemic and national protests against racial injustice [24,25,26,27]. The retail food environment, a social determinant of health, is one setting where consumers interface with an unjust food system [28]. These settings are an integral barrier to FV purchasing and consumption among socially and economically disadvantaged populations [18,29,30]. While large-scale policy and practice efforts have focused on improving dietary quality using food policy, systems, and environmental changes [31], more work is needed to equitably document and dismantle the root causes of FV access and consumption inequities. As the field of nutrition progresses in this new decade, one that remains charged with addressing substantial global food system challenges to meet Sustainable Development Goals [28], it is important we advance scientific knowledge of structural barriers to FV consumption to help achieve food and nutrition security and health equity. Given recent government sector support for food systems solutions (e.g., USDA nutrition security efforts [9], the upcoming White House Conference on Hunger, Nutrition, and Health [32]), the time is now to develop long-term mechanisms that address nutritional inequities [26]. The special issue of the *International Journal of Environmental Research and Public Health*, titled “*Nutrition and Health Equity: Revisiting the Importance of Fruit and Vegetable Availability, Purchasing, and Consumption*”, features several scholarly contributions that emphasize the importance of FV consumption to reduce inequities in health and nutrition. In this commentary, we: (1) summarize current literature on FV accessibility, purchasing, and consumption; (2) draw attention to how special issue publications help to advance food and nutrition security and health equity; and (3) outline key research questions for future investigation from our perspective.

## 2. A Tale of Structural Inequities: Fruit and Vegetable Accessibility and Purchasing

The field has made great strides in documenting geographic gaps in FV accessibility over the past 20 years. The USDA, in particular, provides public data on geographic areas of the United States that lack adequate access to larger retailers that typically offer a large and diverse supply of FVs (i.e., supermarkets, large grocery stores, and supercenters compared to smaller formats such as convenience and drug stores) [33]. It is estimated that 17.4% of Americans, or about 53.6 million people, reside in a U.S. census tract considered to be low-income (≥20% of residents are impoverished) and low-access (a large proportion of residents live more than 1 mile (urban areas) or 10 miles (rural areas) from a supermarket or large grocery store) [34]. A recent USDA report indicated that the number of low-income/low-access census tracts in America had increased slightly from 2010–2015 [35]. They attributed this increase to the Great Recession of the late 2000s, showing how major social and economic crises can negatively affect communities and their socio–environmental attributes [36]. 

This reality is particularly concerning given the long-standing inequities in healthy food accessibility across U.S. social and economic gradients. Studies have shown that communities with large populations of racial/ethnic minorities (particularly non-Hispanic Black and Hispanic populations) and individuals with lower income often have reduced access to food retailers carrying a large supply of affordable, quality, and culturally appropriate FVs in comparison to majority non-Hispanic White and higher income communities [37]. Concurrently, communities of color and communities with lower income tend to have an overabundance of smaller retailers, such as convenience stores, dollar stores, and liquor stores, that stock mostly calorically-dense foods of poor nutritional quality (e.g., candy, sugary beverages, and snack items) and fewer USDA-designated staple foods (fruits, vegetables, bread, meat, and dairy products) [38]. These inequities in accessibility map onto key differences observed in food purchasing practices by race/ethnicity and socioeconomic status among U.S. consumers; low-income individuals and people of color, particularly those who live in low-access communities, continue to face major challenges in procuring FVs [7]. 

Overall, persistent inequities in healthy food accessibility and purchasing have sparked much debate among experts on the terms, definitions, and measures used to evaluate retail food environments [39,40]. Many are moving away from the “food desert” concept (which solely describes poor or absent accessibility to supermarkets and large grocery stores), mainly because interventions aiming to improve FV purchasing and consumption by increasing access to supermarkets/grocery stores have produced inconclusive or null evidence [18,39,40,41], and also because the term inadequately portrays the racist policies and practices that shaped current food environments [42]. Experts have increasingly used the term “food swamp” [43,44]; it describes retail food environments that primarily have smaller retailers in addition to many fast food outlets. Recent evidence suggests that “food swamp” measures may be a better predictor of obesity and poor dietary behaviors than “food desert” measures [43,44]. Additionally, some advocate the use of “food apartheid” when studying inequities in healthy food accessibility [42]; this concept emphasizes the historical significance of racism and racist structures (e.g., segregation, redlining, and disinvestment) to local food systems and healthy food accessibility in communities [42,45]. Although qualitative investigations have described consumer experiences in areas of low food access and “food swamps” [46,47], there continues to be a dearth of quantitative studies that have operationalized and applied this concept to research on healthy food accessibility and purchasing behavior regarding FVs.

Thus, many research topics in the space of FV accessibility and purchasing for food and nutrition security and health equity warrant additional investigation from our perspective, both overall and based on recent events. First, research is needed to understand the impact of the COVID-19 pandemic on U.S. healthy food accessibility and purchasing. Given the number of businesses negatively affected by the pandemic and resulting economic downturn [48], we expect to see significant changes to existing FV accessibility trends and inequities. Second, more research is needed on the role of structural racism, and other systemic forms of oppression, in the perpetuating inequities in FV accessibility and purchasing. Third and lastly, the literature would benefit from more transdisciplinary research that explores connections between social, cultural, and environmental factors (e.g., urban blight, crime, and social disorder) and FV accessibility and purchasing, particularly in communities of color and with lower income. 

## 3. U.S. Fruit and Vegetable Consumption: A Slow Moving Needle

According to the DGA, all Americans should consume an adequate amount and variety of FVs to prevent diet-related chronic disease [3]. Depending on age and sex, a minimum of 1.5–2 cups/day of fruit and 2.5–3 cups/day of vegetables is recommended to maintain a healthy diet [3]. However, the vast majority of American adults and children fall short of these recommendations. Data from the Centers for Disease Control and Prevention estimate less than 12% and 9% of adults met fruit and vegetable recommendations, respectively, in 2015 [49]. Furthermore, data from 2017 suggest less than 7% of adolescents aged 14–18 years old met fruit recommendations, and only 2% met vegetable recommendations [50]. Much of the published trend data suggest that FV consumption rates in the United States have not improved significantly since 2000, although many of the papers on this topic are dated [51,52]. 

FV consumption rates are not equitably distributed across populations; several studies have documented inequities in FV consumption by sex, race/ethnicity, and socioeconomic status [16,53]. National survey data have indicated that females often have higher FV consumption and better overall dietary quality compared to men. Non-Hispanic Black adults, on average, consume fewer servings of vegetables in comparison to non-Hispanic White and Hispanic adults. In addition, FV consumption is often higher among individuals in higher categories of annual household income, poverty level, and educational attainment [16,53]. Trend data demonstrate that these inequities persist [51,52], which further supports the hypothesis that certain marginalized groups continue to face barriers to meeting national recommendations for FV consumption. 

Observational research on inequities in FV consumption has been consistent over the past several decades. However, we believe experts should address several existing research gaps to improve the evidence base on FV consumption. First, current research on inequities in FV consumption by gender identification and sexual orientation is limited [54]. Prior studies evaluated inequities in food security status and overall diet quality, but not FV consumption [55,56]. Second, there continues to be limited intersectional research in this space that highlights inequities within key demographic groups. For example, few studies have evaluated racial/ethnic inequality in FV consumption within socioeconomic classifications [15]. Assari and Lankarani found that higher educational attainment is associated with greater FV consumption among non-Hispanic White adults but not non-Hispanic Black adults [15]. These applications of intersectional theory [57] to observational research on FV consumption may provide new insight into inequities and structural barriers to healthy eating [7]. Third, disaggregated analyses are needed to examine potential heterogeneity in FV consumption within demographic groups. Racial/ethnic minority groups such as Hispanics and Asians have high levels of diversity regarding cultural behavior and practices that influence diet and health. Grouping populations into large categories diminishes our ability to identify specific within-group inequities [58]. And fourth, findings that link structural inequities to individual-level FV consumption are needed to expand the field’s understanding of structural racism and other forms of systemic oppression. 

## 4. Special Issue Contributions to Advancing Nutrition Security and Health Equity

This special issue–*Nutrition and Health Equity: Revisiting the Importance of Fruit and Vegetable Availability, Purchasing, and Consumption*–sought contributions addressing long-standing nutrition and health equity concerns regarding FVs. The seven publications featured in this issue focus on improving purchasing power for FVs among audiences with lower income, mostly through federal nutrition program mechanisms, including the USDA Supplemental Nutrition Assistance Program (SNAP), the Gus Schumacher Nutrition Incentive Program (GusNIP), and the USDA Special Supplemental Nutrition Assistance Program for Women, Infants, and Children (WIC). Several USDA policy efforts situated within the concept of food and nutrition security under the Biden administration prioritize improving factors that influence FV intake beyond individual control. We highlight these strategies and provide a summary of findings regarding special issue publications. 

### 4.1. Thrifty Food Plan

The Thrifty Food Plan (TFP) serves as a basis for determining the financial maximum for SNAP benefits that allow millions of Americans with lower income to purchase food each month [59]. The TFP was updated in 2021 and represented a much-needed increase in SNAP participants’ food purchasing power [60,61]. Furthermore, law (the Agricultural Improvement Act of 2018 (P.L.115–334) or the Farm Bill) now requires the TFP to be revisited every five years. Given higher household expenses from supply chain issues during the COVID-19 pandemic and other global events since early 2020 [62], the U.S. social safety net has proven critical. Without the recent improvements to the TFP, achieving food and nutrition security would be further from reach for many Americans with lower income participating in SNAP. 

Young and Stewart (2022) [63] examined the sufficiency of the TFP increase in SNAP benefits to afford FVs around the country. Importantly, TFP guidance is a standard estimate that does not account for variations in food prices across different U.S. geographies/contexts [60]. While authors found, on average, increased SNAP benefits to be sufficient for households to meet DGA recommendations for FV purchases, affordability barriers by location were evident. Households residing in areas with higher than average food prices likely need to spend a larger percentage of their overall food budget on FVs to meet the DGA relative to households in areas with average or lower than average food prices [63]. As such, this reality likely prevents many Americans with lower incomes who face disproportionate access barriers [64] from purchasing FVs. Authors note the importance of understanding the impact of such policy changes on FV consumption while understanding targeted solutions to mitigate disparities, which deserves more focus in the field [63].

### 4.2. Gus Schumacher Nutrition Incentive Program

The Gus Schumacher Nutrition Incentive Program (GusNIP), formerly called the Food Insecurity Nutrition Incentive Program (FINI), was operationalized in the 2018 Farm Bill and provides competitive grant dollars to organizations administering nutrition incentive or produce prescription programs [65]. Both programs aim to improve FV affordability and thus consumption [66] among Americans with lower income through the provision of a financial incentive or a healthcare prescription used at the point-of-purchase (e.g., farmers markets, retail stores). To evaluate the impact of these efforts on outcomes such as food security and dietary quality (i.e., nutrition security), the GusNIP mechanism also established the Nutrition Incentive Program Training, Technical Assistance, Evaluation, and Information Center (NTAE) [65,67] to help build program capacity at local sites and to evaluate the combined impact of funded projects. In this regard, this funding mechanism is rather novel and has the potential to provide rigorous evidence over time to support food and nutrition policy. 

Budd Nugent et al. (2022) [68] describe in their commentary heterogeneity among nutrition incentive and produce prescription program models and evaluation strategies, which limits opportunities to understand program impact as well as which aspects of program models should be scaled. Budd Nugent and colleagues at the NTAE describe a cooperative approach with researchers, practitioners, and policymakers to establish shared measures to examine evidence from across the United States and among several outcomes (e.g., organizational/program indices, food security, and dietary quality) [68]. The interdisciplinary person power required to achieve such outcomes should be noted, as they describe nuances of defining best practices in data collection among both program types, building local grantee capacity, and providing adequate support to achieve funders’ goals [65,68]. To date, the NTAE has established perhaps the largest compilation resources to support the nutrition incentive and produce prescription fields [67]. Initial evidence from a GusNIP impact evaluation suggests that both programs help to provide the resources necessary to improve food security and dietary quality among Americans with lower incomes [63]. More evidence using shared metrics is needed. 

Further, within the context of GusNIP, rigorous Randomized Control Trials (RCT) are often difficult to employ [68]. Karpyn and colleagues (2022) [69] used a multi-state RCT conducted at farmers markets to test varied nutrition incentive strategies on dietary quality. They found that participants who used a high nutrition incentive (USD 2.00) (compared to moderate and low financial incentives) increased FV consumption by 0.31 cups/day. Higher incentive levels were also associated with more local farmers market spending [69]. These promising results suggest higher nutrition incentive dollar amounts to have a substantial impact on dietary quality among Americans with lower incomes, especially if complemented with local strategies to improve participants’ usage of vouchers [69], which requires future exploration. 

Vargo et al. 2022 [70] focused on nutrition incentive program engagement in Ohio. Among a sample of predominantly female and non-Hispanic Black SNAP participants, authors examined differences between program users and non-users [70]. Several differences among multiple socio–ecological levels were captured. For example, compared to non-users, nutrition incentive program users had higher incomes; larger SNAP budgets (due to household size); were closer in proximity to preferred shopping sites; perceived FV as convenient to access; used a non-grocery site for FV access; used their own car for shopping; had social connections with other program participants; and reported food security. There were also interesting differences between program users who redeemed incentives at a grocery store versus a farmers market, a few being: less time enrolled in the program; lower education; lower frequency of FV purchases; less confidence in FV utilization; and more frequent store trips [70]. Tailored strategies to improve awareness of nutrition incentive programs among Americans with lower income are needed, as non-users reported mostly not knowing about the program. Coordinated capacity building and outreach efforts will also likely need to differ by redemption setting to mitigate disparities in FV purchasing and intake [70].

Lastly, Auvinen and colleagues (2022) [71] conducted interviews among partners and populations relevant to the facilitation of produce prescription programming. While GusNIP currently funds smaller scale produce prescription programs throughout the country [65], these approaches are a growing priority beyond GusNIP, particularly among community organizations and healthcare providers working in high-need areas. Thus, understanding how these interventions might be scalable and sustainable within a healthcare system is warranted. This study outlined several needs to advance this effort: a robust evidence base on the effectiveness of produce prescription programs on client outcomes; strategies to build capacity at multiple levels of program implementation, given produce prescription programs occur across sectors; strategies to reduce implementation cost; and improvements to current healthcare technology (i.e., electronic medical records) to better facilitate produce prescription programming [71]. Many of these noted needs are a priority of the GusNIP NTAE as detailed by Budd et al. (2021), which could serve to inform produce prescription partners beyond the GusNIP funding mechanism [68]. There are opportunities across sectors and funders to engage diverse perspectives to advance the field, including identifying how to best access and use medical record data to build the evidence base for produce prescription programs to improve FV intake and health equity.

### 4.3. Special Supplemental Nutrition Assistance Program for Women, Infants, and Children

The USDA’s WIC aims to improve dietary quality and health among pregnant women and young children (1–5 years) in households with lower income [59]. Allowable program foods were updated in 2009 to better align with the DGA, which resulted in many favorable public health outcomes [72]. A temporary amendment to increase supplemental dollars for WIC households to purchase FVs was also recently prioritized through the American Rescue Plan Act of 2021 (P.L. 117-2) due to the negative impacts of the COVID-19 pandemic on household food budgets [73]. Two qualitative studies published in this special issue explored this change from WIC participant perspectives, both indicating this legislative decision was valued and an important lever to improve FV accessibility among WIC families. 

Duffy et al. (2022) [74] explored awareness, barriers and facilitators, and perceived impact of the higher allotment for FVs among WIC participants in North Carolina. Martinez et al. (2022) [75] explored satisfaction and use of the higher FV allotment among WIC participants in southern California. Results from both studies indicated WIC participants believed the prior allotment for FVs was too low to support accessing adequate variety and quality of FVs [74,75]. Martinez and colleagues also find evidence that the increased FV allotment not only favorability impacted WIC participants, but also improved FV accessibility/consumption for other household members [75]. Duffy and colleagues did find persistent barriers such as social stigma, FV accessibility, and policy dissemination issues described among WIC participants [74]. As such, while the USDA should consider a permanent increase in the dollar allotment for FVs in the WIC benefit package to improve dietary quality [74,75], these key barriers require attention to center equity [74]. 

## 5. Future Directions


*“Food systems enabling fruits and vegetables in healthy diets are not only a technical issue, but bring up very real political, social and ethical questions that societies will have to address, alongside a reliance on evidence [76].”*


According to Kumanyika’s Getting to Equity framework [21], and within the context of FV accessibility, purchasing, and consumption, macro-level changes to policies and systems to improve FV accessibility (availability, transport) and reduce deterrents to purchasing and consumption (e.g., price) are needed. Publications in this special issue help advance knowledge on such aspects and demonstrate the critical role of policy and cross-sector strategies in reducing structural barriers. Other factors such as structurally reinforced discrimination toward racial and ethnic minority groups or local safety concerns [21] in communities likely influence FV accessibility and require more insight. In addition, grassroots strategies to improve FV access, purchasing, and consumption are a critical complement to top-down approaches [21]. Kumanyika notes the importance of improved social and economic resources that might be realized through local policy incentives/disincentives or improving reach of federal nutrition assistance programs, such as SNAP, to eligible, non-participating individuals. Moreover, such work should aim to build community capacity through skills-based support (e.g., to utilize FV), partnerships, and empowerment [21].

Combined with current observational data in the field regarding FV accessibility, purchasing, and consumption, there are several paths forward to realizing food and nutrition security and health equity. Broad recommendations for future research, practice, and policy strategies to improve FV accessibility, purchasing, and consumption, as noted throughout and further detailed below, are summarized in Box 1. For example, advocating for interventions to improve social and economic resources, as proposed in the Getting to Equity Framework [21], is necessary. Broader interventions addressing social and economic disadvantages may be suited to addressing disparities in FV access, purchasing, and consumption because socioeconomic status has been proposed as a fundamental cause of health disparities and a potential mechanism through which structural racism affects health outcomes [77,78,79]. The limited evidence we have on this topic to date has indicated no change in FV consumption after a modest increase in minimum wage [80]. However, given long-standing inequities, more progressive investments to support the health and wealth of socially and economically disadvantaged populations are likely needed. Future work should attempt to link broader poverty mitigation and economic development interventions to outcomes in FV access, purchasing, and consumption, which may increase political will to implement those programs.

Box 1Example Research, Practice, and Policy Directions for Nutrition Security and Health Equity Regarding Fruit and Vegetable (FV) Accessibility, Purchasing, and Consumption.
What are the
impacts of the COVID-19 pandemic, initially and over time, on FV
accessibility, purchasing, and consumption among socially and economically
disadvantaged populations?How does structural racism shape FV accessibility,
purchasing, and consumption among socially and economically disadvantaged
populations?What is the landscape of FV accessibility, purchasing,
and consumption practices by gender identification and sexual orientation?What intersectional factors influence FV accessibility,
purchasing, and consumption among socially and economically disadvantaged
populations?What are the impacts of policies designed to improve FV
accessibility, purchasing, and consumption among socially and economically
disadvantaged populations in various contexts?What strategies improve socially and economically
disadvantaged populations’ knowledge of and engagement with programs designed
to improve FV accessibility, purchasing, and consumption?How do wider social policies (expanding beyond food or
nutrition focus) influence FV accessibility, purchasing, and consumption
among socially and economically disadvantaged populations?Disaggregated data by race/ethnicity, gender
identification, and sexual orientation, for example, are required to
understand experiences with FV accessibility, purchasing, and consumption and
impacts of responsive practice and policy solutions. Transdisciplinary approaches are needed to explore
social, cultural, and environmental linkages with FV accessibility,
purchasing, and consumption among socially and economically disadvantaged
populations.Standardized measures and approaches are warranted, when
possible, across research, practice, and policy investigations to understand
wide-scale impact.Strategies to improve political will to address
structural barriers to FV accessibility, purchasing, and consumption among
socially and economically disadvantaged populations are required. 


Along with broader policy changes, specific interventions tailored to communities that have been marginalized will be necessary to address disparities in FV access, purchasing, and consumption. This is important as color-blind, or race-neutral, policies and interventions may perpetuate existing patterns of inequality [81]. Examples of bottom-up or grassroots strategies may also be more successful than relying on policy solutions given current political will. For example, a Biden Administration and USDA plan to cancel loan debt specifically for minority farmers has been challenged by lawsuits from White farmers claiming racial discrimination [82]. As such, the plan may now exclude many intended recipients [82] and will do less to reduce the long-standing impacts of food apartheid. In accordance with the aforementioned need for better observational data that are disaggregated among racial and ethnic groups [58], the evaluation of policies and interventions intended to serve marginalized populations will require disaggregated outcome indicators by race/ethnicity and include sufficient sample size among subgroups to determine intervention effects [83]. Qualitative and mixed-methods approaches will also be needed in order to elevate and learn from the voices of historically marginalized and resilient communities served by such policies and interventions [84].

## 6. Conclusions

Articles in this special issue provide timely insight for strategies to improve FV accessibility, purchasing, and consumption. Moving forward, multiple overlapping approaches will likely be necessary to address food and nutrition security and health equity. Broad policies to affect socioeconomic differences may address fundamental causes of disparities in FV accessibility, purchasing, and consumption, but targeted interventions to aid communities will also be necessary. The Getting to Equity Framework [21] provides a path forward with example interventions, though the evidence base will need to be improved to advocate for specific interventions that can eliminate FV inequities. Mechanisms to improve FV accessibility and consumption through, or in partnership with, established federal programs such as the SNAP, GusNIP, and WIC are promising. Moving forward, observational and evaluation data, disaggregated and with consideration of intersectional influences, will be required to understand the collective impact of practice and policy interventions that make FV accessible and affordable to all and a regular, acceptable dietary practice for population health and well-being.

## Data Availability

Not applicable.

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
