# Peer review of "Improving Fruit and Vegetable Accessibility, Purchasing, and Consumption to Advance Nutrition Security and Health Equity in the United States"

_ijerph, 2022, doi:10.3390/ijerph191811220_

Round 1

Reviewer 1 Report

Very interesting and relevant comment. The authors carry out a very complete analysis regarding how to improve access, purchase and consumption of fruits and vegetables, in the context of food security. This point is important in the US, but also in other countries, such as Canada, Mexico, France, etc. I only have a few minor comments.

I. Minor Comments:

1. Improve the wording of the objective of the comment.

2. The country (United States) should be included in the title

3. It would be interesting to include a brief paragraph regarding population groups in which increased consumption of fruits and vegetables is a major challenge (example: children and pregnant women)

Author Response

Thank you for the helpful comments. Please see attachment:

Reviewer 2 Report

Thank you for the opportunity to review this interesting work. The authors have summarized the literature on fruits and vegetables (FV) accessibility, purchasing, and consumption. In addition, the authors focused on advances in nutrition security, health equity and outlined key research questions for future investigations.

The authors have done a nice work of summarizing the literature, bringing out potential issues and challenges affecting FV consumption, and provided details on FV consumption in the US and the factors affecting intake. The authors bring out key questions in Table 1 which will help guide the structure of future studies.

In the introduction, the authors talk about a healthy diet and what constitutes a balanced diet and how that can decrease the risk of adverse conditions. It will be helpful if the authors briefly mentioned the dietary guidelines.

Although the main focus of the paper is FV consumption in the US, it will be helpful to have a small section about findings from international studies. FV intake also varies among groups especially among the vulnerable populations such as children. It will be good to have some data on FV intake among children and some differences across populations. The authors do mention how SES and income can affect FV intake but maybe including a small piece about this by providing an international perspective will be helpful. Even if a small paragraph provides data across age groups such as children, women will be helpful.

Lines 154-155: mentions data from the National survey which shows that females have higher FV consumption and better dietary quality compared to men. It will be interesting to see the intake of FV among women from living in other countries especially low-income countries and see if differences are dependent on region, and income level or any other cultural factors. It might be interesting if the authors can shed some light in this area. Some of this information seems to be there in the paper but maybe having it as a separate section will be helpful.

Lines 126-127: is there qualitative data available on healthy food accessibility and purchasing behavior? It might be interesting if some of the future studies in this area consider a mixed methods approach.

Author Response

(The authors gave the same response as above.)

Reviewer 3 Report

It is shocking because this thesis is a direct study of the inequality in FV access according to factors such as income, race, and class that is occurring in the United States, and inequality in health products.

In addition, there is a lot of agreement that the ongoing COVID-19 would have worsened FV access and purchasing access that would have worsened further. And I fully agree with the prediction that Fruit and Vegetable Consumption in the U.S. will act as the nation's Achilles heel in the future. I would like to express my sincere thanks to the papers that are not prior, but are making efforts to improve and publish a report on the universal dignity of the human race for the improvement of the future society. In addition, the spread of inequality in FV access and purchasing access not only to the United States but also to underdeveloped countries and other countries would be something to be wary of. We confirmed that the financial crisis in developed countries that occurred in 2008 was spreading to the world, and also confirmed how terrifying the results were for the entire human society. I think that it is necessary to urgently research and discuss these issues in depth and spread them to resolve conflicts, resolve FV and purchase access, and strive for a complete solution through the intervention of governments and international organizations.

Finally, it is hoped that the verification process of various problems presented to resolve FV consumption inequality should be continuously checked and secured. It is thought that there is a universal operating principle as explained in this thesis to expand and implement the overall food, health, and administrative safety net not only for Americans but also for the entire human race. Programs like TFP are excellent programs and we expect them to grow as a safety net through continuous improvement.

Author Response

(The authors gave the same response as above.)
